# Unsupervised Learning of Shape and Pose with Differentiable Point Clouds

**Eldar Insafutdinov**[*]
Max Planck Institute for Informatics
eldar@mpi-inf.mpg.de

**Alexey Dosovitskiy**
Intel Labs
adosovitskiy@gmail.com

## Abstract

We address the problem of learning accurate 3D shape and camera pose from a collection of unlabeled category-specific images. We train a convolutional network to predict both the shape and the pose from a single image by minimizing the reprojection error: given several views of an object, the projections of the predicted shapes to the predicted camera poses should match the provided views. To deal with pose ambiguity, we introduce an ensemble of pose predictors which we then distill to a single "student" model. To allow for efficient learning of high-fidelity shapes, we represent the shapes by point clouds and devise a formulation allowing for differentiable projection of these. Our experiments show that the distilled ensemble of pose predictors learns to estimate the pose accurately, while the point cloud representation allows to predict detailed shape models.

## 1  Introduction

We live in a three-dimensional world, and a proper understanding of its volumetric structure is crucial for acting and planning. However, we perceive the world mainly via its two-dimensional projections. Based on these projections, we are able to infer the three-dimensional shapes and poses of the surrounding objects. How does this volumetric shape perception emerge from observing only from two-dimensional projections? Is it possible to design learning systems with similar capabilities?

Deep learning methods have recently shown promise in addressing these questions [25, 20]. Given a set of views of an object and the corresponding camera poses, these methods learn 3D shape via the reprojection error: given an estimated shape, one can project it to the known camera views and compare to the provided images. The discrepancy between these generated projections and the training samples provides training signal for improving the shape estimate. Existing methods of this type have two general restrictions. First, these approaches assume that the camera poses are known precisely for all provided images. This is a practically and biologically unrealistic assumption: a typical intelligent agent only has access to its observations, not its precise location relative to objects in the world. Second, the shape is predicted as a low-resolution (usually $32^3$ voxels) voxelated volume. This representation can only describe very rough shape of an object. It should be possible to learn finer shape details from 2D supervision.

In this paper, we learn high-fidelity shape models solely from their projections, without ground truth camera poses. This setup is challenging for two reasons. First, estimating both shape and pose is a chicken-and-egg problem: without a good shape estimate it is impossible to learn accurate pose because the projections would be uninformative, and vice versa, an accurate pose estimate is necessary to learn the shape. Second, pose estimation is prone to local minima caused by ambiguity: an object may look similar from two viewpoints, and if the network converges to predicting only one of these in all cases, it will not be able to learn predicting the other one. We find that the first problem can be solved surprisingly well by joint optimization of shape and pose predictors: in practice, good

---

[*]Work done while interning at Intel.

shape estimates can be learned even with relatively noisy pose predictions. The second problem, however, leads to drastic errors in pose estimation. To address this, we train a diverse ensemble of pose predictors and distill those to a single student model.

To allow learning of high-fidelity shapes, we use the point cloud representation, in contrast with voxels used in previous works. Point clouds allow for computationally efficient processing, can produce high-quality shape models [6], and are conceptually attractive because they can be seen as "matter-centric", as opposed to "space-centric" voxel grids. To enable learning point clouds without explicit 3D supervision, we implement a differentiable projection operator that, given a point set and a camera pose, generates a 2D projection – a silhouette, a color image, or a depth map. We dub the formulation "Differentiable Point Clouds".

We evaluate the proposed approach on the task of estimating the shape and the camera pose from a single image of an object [2]. The method successfully learns to predict both the shape and the pose, with only a minor performance drop relative to a model trained with ground truth camera poses. The point-cloud-based formulation allows for effective learning of high-fidelity shape models when provided with images of sufficiently high resolution as supervision. We demonstrate learning point clouds from silhouettes and augmenting those with color if color images are available during training. Finally, we show how the point cloud representation allows to automatically discover semantic correspondences between objects.

## 2   Related Work

Reconstruction of three-dimensional shapes from their two-dimensional projections has a long history in computer vision, constituting the field of 3D reconstruction. A review of this field goes outside of the scope of this paper; however, we briefly list several related methods. Cashman and Fitzgibbon [2] use silhouettes and keypoint annotation to reconstruct deformable shape models from small class-specific image collections, Vicente et al. [22] apply similar methods to a large-scale Pascal VOC dataset, Tulsiani et al. [18] reduce required supervision by leveraging computer vision techniques. These methods show impressive results even in the small data regime; however, they have difficulties with representing diverse and complex shapes. Loper and Black [12] implement a differentiable renderer and apply it for analysis-by-synthesis. Our work is similar in spirit, but operates on point clouds and integrates the idea of differentiable rendering with deep learning. The approach of Rhodin et al. [14] is similar to our technically in that it models human body with a set of Gaussian density functions and renders them using a physics-motivated equation for light transport. Unlike in our approach, the representation is not integrated into the learning framework and requires careful initial placement of the Gaussians, making it unsuitable for automated reconstruction of arbitrary shape categories. Moreover, the projection method scales quadratically with the number of Gaussians, which limits the maximum fidelity of the shapes being represented.

Recently the task of learning 3D structure from 2D supervision is being addressed with deep-learning-based methods. The methods are typically based on reprojection error – comparing 2D projections of a predicted 3D shape to the ground truth 2D projections. Yan et al. [25] learn 3D shape from silhouettes, via a projection operation based on selecting the maximum occupancy value along a ray. Tulsiani et al. [20] devise a differentiable formulation based on ray collision probabilities and apply it to learning from silhouettes, depth maps, color images, and semantic segmentation maps. Lin et al. [11] represent point clouds by depth maps and re-project them using a high resolution grid and inverse depth max-pooling. Concurrently with us, Kato et al. [8] propose a differentiable renderer for meshes and use it for learning mesh-based representations of object shapes. All these methods require exact ground truth camera pose corresponding to the 2D projections used for training. In contrast, we aim to relax this unrealistic assumption and learn only from the projections.

Rezende et al. [13] explore several approaches to generative modeling of 3D shapes based on their 2D views. One of the approaches does not require the knowledge of ground truth camera pose; however, it is only demonstrated on a simple dataset of textured geometric primitives. Most related to our submission is the concurrent work of Tulsiani et al. [21]. The work extends the Differentiable Ray Consistency formulation [20] to learning without pose supervision. The method is voxel-based and deals with the complications of unsupervised pose learning using reinforcement learning and

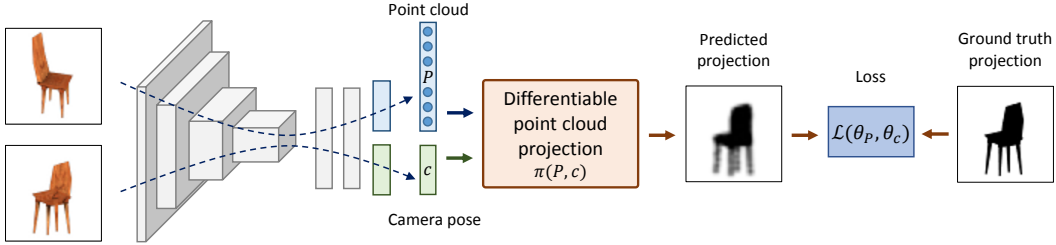

Figure 1: Learning to predict the shape and the camera pose. Given two views of the same object, we predict the corresponding shape (represented as a point cloud) and the camera pose. Then we use a differentiable projection module to generate the view of the predicted shape from the predicted camera pose. Dissimilarity between this synthesized projection and the ground truth view serves as the training signal.

a GAN-based prior. In contrast, we make use of a point cloud representation, use an ensemble to predict the pose, and do not require a prior on the camera poses.

The issue of representation is central to deep learning with volumetric data. The most commonly used structure is a voxel grid - a direct 3D counterpart of a 2D pixelated image [5, 23]. This similarity allows for simple transfer of convolutional network architectures from 2D to 3D. However, on the downside, the voxel grid representation leads to memory- and computation-hungry architectures. This motivates the search for alternative options. Existing solutions include octrees [17], meshes [8, 26], part-based representations [19, 10], multi-view depth maps [15], object skeletons [24], and point clouds [6, 11]. We choose to use point clouds in this work, since they are less overcomplete than voxel grids and allow for effective networks architectures, but at the same time are more flexible than mesh-based or skeleton-based representations.

## 3 Single-view Shape and Pose Estimation

We address the task of predicting the three-dimensional shape of an object and the camera pose from a single view of the object. Assume we are given a dataset $D$ of views of $K$ objects, with $m_i$ views available for the $i$-th object: $D = \cup_{i=1}^{K}\{\langle \mathbf{x}_j^i, \mathbf{p}_j^i \rangle\}_{j=1}^{m_i}$. Here $\mathbf{x}_j^i$ denotes a color image and $\mathbf{p}_j^i$ – the projection of some modality (silhouette, depth map of a color image) from the same view. Each view may be accompanied with the corresponding camera pose $c_j^i$, but the more interesting case is when the camera poses are not known. We focus on this more difficult scenario in the remainder of this section.

An overview of the model is shown in Figure 1. Assume we are given two images $\mathbf{x}_1$ and $\mathbf{x}_2$ of the same object. We use parametric function approximators to predict a 3D shape (represented by a point cloud) from one of them $\hat{P}_1 = F_P(\mathbf{x}_1, \theta_P)$, and the camera pose from the other one: $\hat{c}_2 = F_c(\mathbf{x}_2, \theta_c)$. In our case, $F_P$ and $F_c$ are convolutional networks that share most of their parameters. Both the shape and the pose are predicted as fixed-length vectors using fully connected layers.

Given the predictions, we render the predicted shape from the predicted view: $\hat{\mathbf{p}}_{1,2} = \pi(\hat{P}_1, \hat{c}_2)$, where $\pi$ denotes the differentiable point cloud renderer described in Section 4. The loss function is then the discrepancy between this predicted projection and the ground truth. We use standard MSE in this work both for all modalities, summed over the whole dataset:

$$\mathcal{L}(\theta_P, \theta_c) = \sum_{i=1}^{N} \sum_{j_1, j_2=1}^{m_i} \left\| \hat{\mathbf{p}}_{j_1, j_2}^i - \mathbf{p}_{j_2}^i \right\|^2. \tag{1}$$

Intuitively, this training procedure requires that for all pairs of views of the same object, the renderings of the predicted point cloud match the provided ground truth views.

**Estimating pose with a distilled ensemble.** We found that the basic implementation described above fails to predict accurate poses. This is caused by local minima: the pose predictor converges to either estimating all objects as viewed from the back, or all viewed from the front. Indeed, based on silhouettes, it is difficult to distinguish between certain views even for a human, see Figure 2 (a).

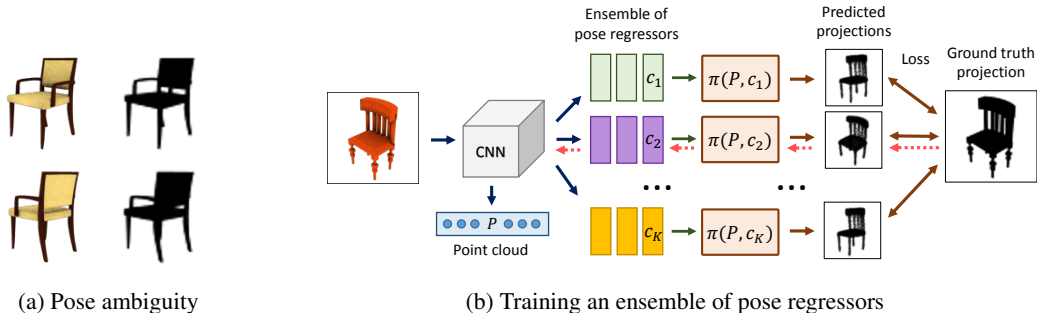

(a) Pose ambiguity                    (b) Training an ensemble of pose regressors

Figure 2: (a) Pose ambiguity: segmentation masks, which we use for supervision, look very similar from different camera views. (b) The proposed ensemble of pose regressors designed to resolve this ambiguity. The network predicts a diverse set $\{c_k\}_{k=1}^{K}$ of pose candidates, each of which is used to compute a projection of the predicted point cloud $P$. The weight update (backward pass shown in dashed red) is only performed for the pose candidate yielding the projection that best matches the ground truth.

To alleviate this issue, instead of a single pose regressor $F_c(\cdot, \theta_c)$, we introduce an ensemble of $K$ pose regressors $F_c^k(\cdot, \theta_c^k)$ (see Figure 2 (b)) and train the system with the "hindsight" loss [7, 4]:

$$\mathcal{L}_h(\theta_P, \theta_c^1, \ldots, \theta_c^K) = \min_{k \in [1,K]} \mathcal{L}(\theta_P, \theta_c^k). \tag{2}$$

The idea is that each of the predictors learns to specialize on a subset of poses and together they cover the whole range of possible values. No special measures are needed to ensure this specialization: it emerges naturally as a result of random weight initialization if the network architecture is appropriate. Namely, the different pose predictors need to have several (at least 3, in our experience) non-shared layers.

In parallel with training the ensemble, we distill it to a single regressor by using the best model from the ensemble as the teacher. This best model is selected based on the loss, as in Eq. (2). At test time we discard the ensemble and use the distilled regressor to estimate the camera pose. The loss for training the student is computed as an angular difference between two rotations represented by quaternions: $L(q_1, q_2) = 1 - \mathrm{Re}(q_1 q_2^{-1} / \|q_1 q_2^{-1}\|)$, where $\mathrm{Re}$ denotes the real part of the quaternion. We found that standard MSE loss performs poorly when regressing rotation.

**Network architecture.** We implement the shape and pose predictor with a convolutional network with two branches. The network starts with a convolutional encoder with a total of 7 layers, 4 of which have stride 2. These are followed by 2 shared fully connected layers, after which the network splits into two branches for shape and pose prediction. The shape branch is an MLP with one hidden layer. The point cloud of $N$ points is predicted as a vector with dimensionality $3N$ (point positions) or $6N$ (positions and RGB values). The pose branch is an MLP with one shared hidden layer and two more hidden layers for each of the pose predictors. The camera pose is predicted as a quaternion. In the ensemble model we use $K = 4$ pose predictors. The "student" model is another branch with the same architecture.

## 4 Differentiable Point Clouds

A key component of our model is the differentiable point cloud renderer $\pi$. Given a point cloud $P$ and a camera pose $c$, it generates a view $\mathbf{p} = \pi(P, c)$. The point cloud may have a signal, such as color, associated with it, in which case the signal can be projected to the view.

The high-level idea of the method is to smooth the point cloud by representing the points with density functions. Formally, we assume the point cloud is a set of $N$ tuples $P = \{\langle \mathbf{x}_i, \mathbf{s}_i, \mathbf{y}_i \rangle\}_{i=1}^{N}$, each including the point position $\mathbf{x}_i = (x_{i,1}, x_{i,2}, x_{i,3})$, the size parameter $\mathbf{s}_i$, and the associated signal $\mathbf{y}_i$ (for instance, an RGB color). In most of our experiments the size parameter is a two-dimensional vector including the covariance of an isotropic Gaussian and a scaling factor. However, in general $\mathbf{s}_i$ can represent an arbitrary parametric distribution: for instance, in the supplement we

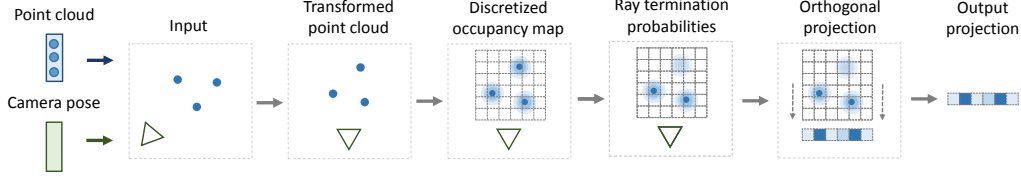

Figure 3: Differentiable rendering of a point cloud. We show 2D-to-1D projection for illustration purposes, but in practice we perform 3D-to-2D projection. The points are transformed according to the camera parameters, smoothed, and discretized. We perform occlusion reasoning via a form of ray tracing, and finally project the result orthogonally.

show experiments with Gaussians with a full covariance matrix. The size parameters can be either specified manually or learned jointly with the point positions.

The overall differentiable rendering pipeline is illustrated in Figure 3. For illustration purposes we show 2D-to-1D projection in the figure, but in practice we perform 3D-to-2D projection. We start by transforming the positions of points to the standard coordinate frame by the projective transformation $T_c$ corresponding to the camera pose $c$ of interest: $\mathbf{x}'_i = T_c \mathbf{x}_i$. The transform $T_c$ accounts for both extrinsic and intrinsic camera parameters. We also compute the transformed size parameters $\mathbf{s}'$ (the exact transformation rule depends on the distribution used). We set up the camera transformation matrix such that after the transform, the projection amounts to orthogonal projection along the third axis.

To allow for the gradient flow, we represent each point $\langle \mathbf{x}_i, \mathbf{s}_i \rangle$ by a smooth function $f_i(\cdot)$. In this work we set $f_i$ to scaled Gaussian densities. The occupancy function of the point cloud is a clipped sum of the individual per-point functions:

$$o(\mathbf{x}) = \mathrm{clip}(\sum_{i=1}^{N} f_i(\mathbf{x}), [0,1]), \qquad f_i(\mathbf{x}) = c_i \exp\left(-\frac{1}{2}(\mathbf{x} - \mathbf{x}'_i)^T \Sigma_i^{-1}(\mathbf{x} - \mathbf{x}'_i)\right), \qquad (3)$$

where $\langle c_i, \Sigma_i \rangle = \mathbf{s}_i$ are the size parameters. We discretize the resulting function to a grid of resolution $D_1 \times D_2 \times D_3$. Note that the third index corresponds to the projection axis, with index 1 being the closest to the camera and $D_3$ – the furthest from the camera.

Before projecting the resulting volume to a plane, we need to ensure that the signal from the occluded points does not interfere with the foreground points. To this end, we perform occlusion reasoning using a differentiable ray tracing formulation, similar to Tulsiani et al. [20]. We convert the occupancies $o$ to ray termination probabilities $r$ as follows:

$$r_{k_1,k_2,k_3} = o_{k_1,k_2,k_3} \prod_{u=1}^{k_3-1}(1 - o_{k_1,k_2,u}) \ \ if \ \ k_3 \leqslant D_3, \quad r_{k_1,k_2,D_3+1} = \prod_{u=1}^{D_3}(1 - o_{k_1,k_2,u}). \quad (4)$$

Intuitively, a cell has high termination probability $r_{k_1,k_2,k_3}$ if its occupancy value $o_{k_1,k_2,k_3}$ is high and all previous occupancy values $\{o_{k_1,k_2,u}\}_{u<k_3}$ are low. The additional background cell $r_{k_1,k_2,D_3+1}$ serves to ensure that the termination probabilities sum to 1.

Finally, we project the volume to the plane:

$$p_{k_1,k_2} = \sum_{k_3=1}^{D_3+1} r_{k_1,k_2,k_3} y_{k_1,k_2,k_3}. \qquad (5)$$

Here $y$ is the signal being projected, which defines the modality of the result. To obtain a silhouette, we set $y_{k_1,k_2,k_3} = 1 - \delta_{k_3,D_3+1}$. For a depth map, we set $y_{k_1,k_2,k_3} = k_3/D_3$. Finally, to project a signal $\mathbf{y}$ associated with the point cloud, such as color, we set $y$ to a discretized version of the normalized signal distribution: $\mathbf{y}(\mathbf{x}) = \sum_{i=1}^{N} \mathbf{y}_i f_i(\mathbf{x}) / \sum_{i=1}^{N} f_i(\mathbf{x})$.

## 4.1 Implementation details

Technically, the most complex part of the algorithm is the conversion of a point cloud to a volume. We have experimented with two implementations of this step: one that is simple and flexible (we

refer to it as `basic`) and another version that is less flexible, but much more efficient (we refer to it as `fast`). We implemented both versions using standard Tensorflow [1] operations. At a high level, in the `basic` implementation each function $f_i$ is computed on an individual volumetric grid, and the results are summed. This allows for flexibility in the choice of the function class, but leads to both computational and memory requirements growing linearly with both the number of points $N$ and the volume of the grid $V$, resulting in the complexity $O(NV)$. The `fast` version scales more gracefully, as $O(N + V)$. This comes at the cost of using the same kernel for all functions $f_i$. The `fast` implementation performs the operation in two steps: first putting all points on the grid with trilinear interpolation, then applying a convolution with the kernel. Further details are provided in the supplement.

## 5 Experiments

### 5.1 Experimental setup

**Datasets.** We conduct the experiments on 3D models from the ShapeNet [3] dataset. We focus on 3 categories typically used in related work: chairs, cars, and airplanes. We follow the train/test protocol and the data generation procedure of Tulsiani et al. [20]: split the models into training, validation and test sets and render 5 random views of each model with random light source positions and random camera azimuth and elevation, sampled uniformly from $[0°, 360°)$ and $[-20°, 40°]$ respectively.

**Evaluation metrics.** We use the Chamfer distance as our main evaluation metric, since it has been shown to be well correlated with human judgment of shape similarity [16]. Given a ground truth point cloud $P^{gt} = \{\mathbf{x}_n^{gt}\}$ and a predicted point cloud $P^{pr} = \{\mathbf{x}_n^{pr}\}$, the distance is defined as follows:

$$d_{Chamf}(P^{gt}, P^{pred}) = \frac{1}{|P^{pr}|} \sum_{\mathbf{x}^{pr} \in P^{pr}} \min_{\mathbf{x} \in P^{gt}} \|\mathbf{x}^{pr} - \mathbf{x}\|_2 + \frac{1}{|P^{gt}|} \sum_{\mathbf{x}^{gt} \in P^{gt}} \min_{\mathbf{x} \in P^{pr}} \|\mathbf{x}^{gt} - \mathbf{x}\|_2 . \quad (6)$$

The two sums in Eq. (6) have clear intuitive meanings. The first sum evaluates the precision of the predicted point cloud by computing how far on average is the closest ground truth point from a predicted point. The second sum measures the coverage of the ground truth by the predicted point cloud: how far is on average the closest predicted point from a ground truth point.

For measuring the pose error, we use the same metrics as Tulsiani et al. [21]: accuracy (the percentage of samples for which the predicted pose is within $30°$ of the ground truth) and the median error (in degrees). Before starting the pose and shape evaluation, we align the canonical pose learned by the network with the canonical pose in the dataset, using Iterative Closest Point (ICP) algorithm on the first 20 models in the validation set. Further details are provided in the supplement.

**Training details.** We trained the networks using the Adam optimizer [9], for 600,000 mini-batch iterations. We used mini-batches of 16 samples (4 views of 4 objects). We used a fixed learning rate of 0.0001 and the standard momentum parameters. We used the `fast` projection in most experiments, unless mentioned otherwise. We varied both the number of points in the point cloud and the resolution of the volume used in the projection operation depending on the resolution of the ground truth projections used for supervision. We used the volume with the same side as the training samples (e.g., $64^3$ volume for $64^2$ projections), and we used 2000 points for $32^2$ projections, 8000 points for $64^2$ projections, and 16,000 points for $128^2$ projections.

When predicting dense point clouds, we have found it useful to apply dropout to the predictions of the network to ensure even distribution of points on the shape. Dropout effects in selecting only a subset of all predicted points for projection and loss computation. In experiments reported in Sections 5.2 and 5.3 we started with a very high $90\%$ dropout and linearly reduced it to 0 towards the end of training. We also implemented a schedule for the point size parameters, linearly decreasing from $5\%$ of the projection volume size to $0.3\%$ over the course of training. The scaling coefficient of the points was learned in all experiments. An ablation study is shown in the supplement.

**Computational efficiency.** A practical advantage of a point-cloud-based method is that it does not require using a 3D convolutional decoder as required by voxel-based methods. This improves the efficiency and allows the method to better scale to higher resolution. For resolution 32 the training times of the methods are roughly on par. For 64 the training time of our method is roughly 1 day in contrast with 2.5 days for its voxel-based counterpart. For 128 the training time of our method is 3 days, while the voxel-based method does not fit into 12Gb of GPU memory with our batch size.

|           | Resolution 32 |         |        |      | Resolution 64 |      | Resolution 128 |      |
|-----------|---------------|---------|--------|------|---------------|------|----------------|------|
|           | DRC [20]      | PTN [25]| Ours-V | Ours | Ours-V        | Ours | EPCG [11]      | Ours |
| Airplane  | 8.35          | 3.79    | 5.57   | 4.52 | 4.94          | 3.50 | 4.03           | **2.84** |
| Car       | 4.35          | 3.94    | 3.88   | 4.22 | 3.41          | 2.98 | 3.69           | **2.42** |
| Chair     | 8.01          | 5.10    | 5.57   | 5.10 | 4.80          | 4.15 | 5.62           | **3.62** |
| Mean      | 6.90          | 4.27    | 5.01   | 4.61 | 4.39          | 3.55 | 4.45           | **2.96** |

Table 1: Quantitative results on shape prediction with known camera pose. We report the Chamfer distance between normalized point clouds, multiplied by 100. Our point-cloud-based method (Ours) outperforms its voxel-based counterpart (Ours-V) and benefits from higher resolution training samples.

| Input | View 1 | View 2 | Input | View 1 | View 2 | Input | View 1 | View 2 |
|-------|--------|--------|-------|--------|--------|-------|--------|--------|

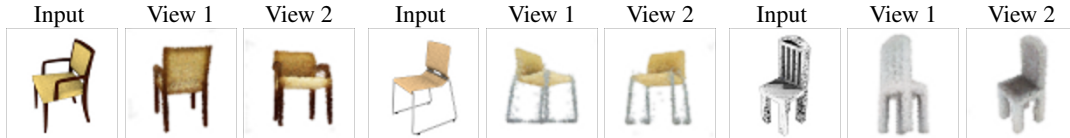

Figure 4: Learning colored point clouds. Best viewed on screen. We show the input image, as well as two renderings of the predicted point cloud from other views. The general color is preserved well, but the fine details may be lost.

## 5.2 Estimating shape with known pose

**Comparison with baselines.** We start by benchmarking the proposed formulation against existing methods in the simple setup with known ground truth camera poses and silhouette-based training. We compare to Perspective Transformer Networks (PTN) of Yan et al. [25], Differentiable Ray Consistency (DRC) of Tulsiani et al. [20], Efficient Point Cloud Generation (EPCG) of Lin et al. [11], and to the voxel-based counterpart of our method. PTN and DRC are only available for $32^3$ output voxel grid resolution. EPCG uses the point cloud representation, same as our method. However, in the original work EPCG has only been evaluated in the unrealistic setup of having 100 random views per object and pre-training from 8 fixed views (corners of a cube). We re-train this method in the more realistic setting used in this work – 5 random views per object.

The quantitative results are shown in Table 1. Our point-cloud-based formulation (Ours) outperforms its voxel-based counterpart (Ours-V) in all cases. It improves when provided with high resolution training signal, and benefits from it more than the voxel-based method. Overall, our best model (at 128 resolution) decreases the mean error by 30% compared to the best baseline. An interesting observation is that at low resolution, PTN performs remarkably well, closely followed by our point-cloud-based formulation. Note, however, that the PTN formulation only applies to learning from silhouettes and cannot be easily generalized to other modalities.

Our model achieves 50% improvement over the point cloud method EPCG, despite it being trained from depth maps, which is a stronger supervision compared to silhouettes used for our models. When trained with silhouette supervision only, EPCG achieves an average error of 8.20, 2.7 times worse than our model. We believe our model is more successful because our rendering procedure is differentiable w.r.t. all three coordinates of points, while the method of Lin et al. – only w.r.t. the depth.

**Colored point clouds.** Our formulation supports training with other supervision than silhouettes, for instance, color. In Figure 4 we demonstrate qualitative results of learning colored point clouds with our method. Despite challenges presented by the variation in lighting and shading between different views, the method is able to learn correctly colored point clouds. For objects with complex textures the predicted colors get blurred (last example).

**Learnable covariance.** In the experiments reported above we have learnt point clouds with all points having identical isotropic covariance matrices. We conducted additional experiments where covariance matrices are learnt jointly with point positions, allowing for more flexible representation of shapes. Results are reported in the supplement.

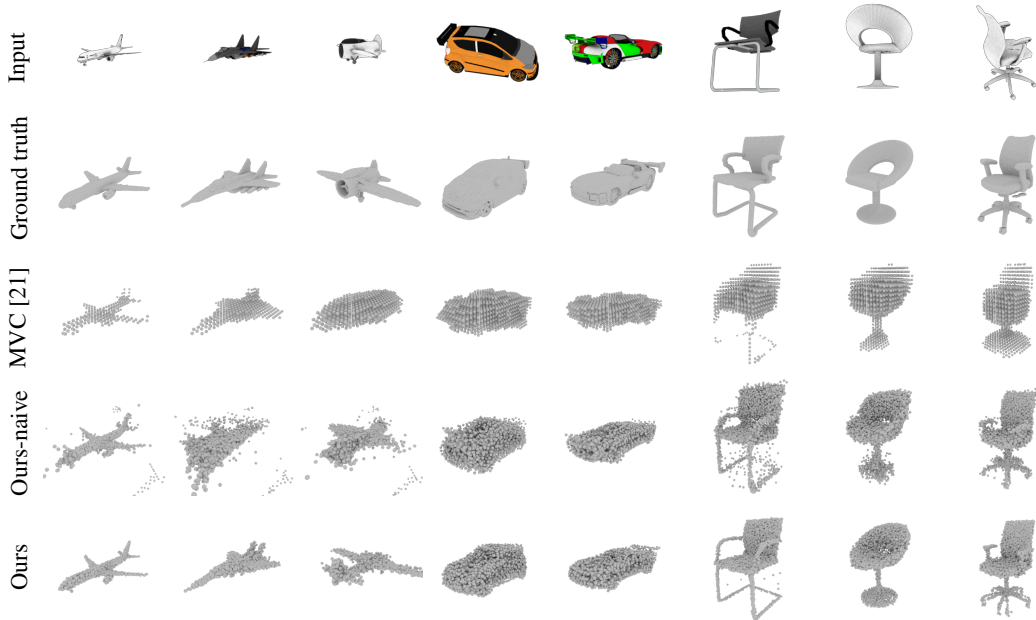

Figure 5: Qualitative results of shape prediction. Best viewed on screen. Shapes predicted by our naive model with a single pose predictor (Ours-naive) are more detailed than those of MVC [21]. The model with an ensemble of pose predictors (Ours) generates yet sharper shapes. The point cloud representation allows to preserve fine details such as thin chair legs.

|  | Shape ($D_{Chamf}$) | | | Pose (Accuracy & Median error) | | | | | | | |
|---|---|---|---|---|---|---|---|---|---|---|---|
|  | MVC [21] | Ours-naive | Ours | GT pose [21] | | MVC [21] | | Ours-naive | | Ours | |
| Airplane | 4.43 | 7.22 | **3.91** | **0.79** | 10.7 | 0.69 | 14.3 | 0.20 | 100.2 | 0.75 | **8.2** |
| Car | 4.16 | 4.14 | **3.47** | **0.90** | 7.4 | 0.87 | 5.2 | 0.49 | 42.8 | 0.86 | **5.0** |
| Chair | 6.51 | 4.79 | **4.30** | 0.85 | 11.2 | 0.81 | **7.8** | 0.50 | 31.3 | **0.86** | 8.1 |
| Mean | 5.04 | 5.38 | **3.89** | **0.85** | 10.0 | 0.79 | 9.0 | 0.40 | 58.1 | 0.82 | **7.1** |

Table 2: Quantitative results of shape and pose prediction. Best results for each metric are highlighted in bold. The naive version of our method predicts the shape quite well, but fails to predict accurate pose. The full version predicts both shape and pose well.

## 5.3 Estimating shape and pose

We now drop the unrealistic assumption of having the ground truth camera pose during training and experiment with predicting both the shape and the camera pose. We use the ground truth at 64 pixel resolution for our method in these experiments. We compare to the concurrent Multi-View Consistency (MVC) approach of Tulsiani et al. [21], using results reported by the authors for pose estimation and pre-trained models provided by the authors for shape evaluations.

Quantitative results are provided in Table 2. Our naive model (Ours-naive) learns quite accurate shape (7% worse than MVC), despite not being able to predict the pose well. Our explanation is that predicting wrong pose for similarly looking projections does not significantly hamper the training of the shape predictor. Shape predicted by the full model (Ours) is yet more precise: 28% more accurate than MVC and only 10% less accurate than with ground truth pose (as reported in Table 1). Pose prediction improves dramatically, thanks to the diverse ensemble formulation. As a result, our pose prediction results are on average slightly better than those of MVC [21] in both metrics, and even better in median error than the results of training with ground truth pose labels (as reported by Tulsiani et al. [21]).

Figure 5 shows a qualitative comparison of shapes generated with different methods. Even the results of the naive model (Ours-naive) compare favorably to MVC [21]. Introducing the pose ensemble

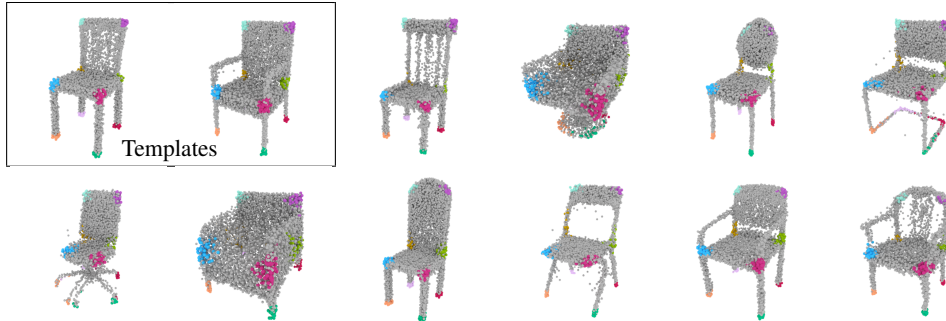

Figure 6: Discovered semantic correspondences. Points of the same color correspond to the same subset in the point cloud across different instances. The points were selected on two template instances (top left). Best viewed on screen.

leads to learning more accurate pose and, as a consequence, more precise shapes. These results demonstrate the advantage of the point cloud representation over the voxel-based one. Point clouds are especially suitable for representing fine details, such as thin legs of the chairs. We also show typical failure cases of the proposed method. One of the airplanes is rotated by $180$ degrees, since the network does not have a way to find which orientation is considered correct. The shapes of two of the chairs somewhat differ from the true shapes. This is because of the complexity of the training problem and, possibly, overfitting. Yet, the shapes look detailed and realistic.

## 5.4 Discovery of semantic correspondences

Besides higher shape fidelity, the "matter-centric" point cloud representation has another advantage over the "space-centric" voxel representation: there is a natural correspondence between points in different predicted point clouds. Since we predict points with a fully connected layer, the points generated by the same output unit in different shapes can be expected to carry similar semantic meaning. We empirically verify this hypothesis. We choose two instances from the validation set of the chair category as templates (shown in the top-left corner of Figure 6) and manually annotate 3D keypoint locations corresponding to characteristic parts, such as corners of the seat, tips of the legs, etc. Then, for each keypoint we select all points in the predicted clouds within a small distance from the keypoint and compute the intersection of the points indices between the two templates. (Intersection of indices between two object instances is not strictly necessary, but we found it to slightly improve the quality of the resulting correspondences.) We then visualize points with these indices on several other object instances, highlighting each set of points with a different color. Results are shown in Figure 6. As hypothesized, selected points tend to represent the same object parts in different object instances. Note that no explicit supervision was imposed towards this goal: semantic correspondences emerge automatically. We attribute this to the implicit ability of the model to learn a regular, smooth representation of the output shape space, which is facilitated by reusing the same points for the same object parts.

## 6 Conclusion

We have proposed a method for learning pose and shape of 3D objects given only their 2D projections, using the point cloud representation. Extensive validation has shown that point clouds compare favorably with the voxel-based representation in terms of efficiency and accuracy. Our work opens up multiple avenues for future research. First, our projection method requires an explicit volume to perform occlusion reasoning. We believe this is just an implementation detail, which might be relaxed in the future with a custom rendering procedure. Second, since the method does not require accurate ground truth camera poses, it could be applied to learning from real-world data. Learning from color images or videos would be especially exciting, but it would require explicit reasoning about lighting and shading, as well as dealing with the background. Third, we used a very basic decoder architecture for generating point clouds, and we believe more advanced architectures [26] could improve both the efficiency and the accuracy of the method. Finally, the fact that the loss is explicitly computed on projections (in contrast with, e.g., Tulsiani et al. [20]), allows directly applying advanced techniques from the 2D domain, such as perceptual losses and GANs, to learning 3D representations.

**Acknowledgements**

We would like to thank René Ranftl and Stephan Richter for valuable discussions and feedback. We would also like to thank Shubham Tulsiani for providing the models of the MVC method for testing.

## Footnotes

[2]The project website with code can be found at `https://eldar.github.io/PointClouds/`.

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
