[Supplementary Material]

# Supplementary material for "Unsupervised Learning of Shape and Pose with Differentiable Point Clouds"

**Eldar Insafutdinov**[*]
Max Planck Institute for Informatics
eldar@mpi-inf.mpg.de

**Alexey Dosovitskiy**
Intel Labs
adosovitskiy@gmail.com

## A  Implementation details

### A.1  Network architecture

The convolutional encoder includes 7 layers. The first one has a $5 \times 5$ kernel with 16 channels and stride 2. The remaining layers all have 3 kernels and come in pairs. The first layer in the pair has stride 2, the second one – stride 1. The number of channels grows by a factor of 2 after each strided layer. The convolutional encoder is followed by two fully connected layers with 1024 units. Then the network separates into two branches predicting shape and pose. The shape branch has one hidden layer with 1024 units and then predicts the point cloud. The pose branch has one shared hidden layer with 1024 units. In the naive variant of the method, pose is predicted directly from this hidden layer. In the full approach with an ensemble of pose predictors, this layer is followed by 2 separate hidden layers for each pose predictor in the ensemble, with 32 units each. We used leaky ReLU with the negative slope 0.2 after all layers except for the shape prediction layer where we used the $tanh$ non-linearity to constrain the output coordinates.

### A.2  Differentiable point cloud projection

Assume we are given a set of $N$ points with coordinates and sizes $\{(\mathbf{x}_n, \sigma_n)\}_{n=0}^{N-1}$, as well as the desired spatial dimensions $D_1 \times D_2 \times D_3$ of the volume to be used for projection. Here we assume indexing of all tensors is 0-based.

In the `basic` implementation, we start by creating a coordinate tensor $\mathbf{M}$ of dimensions $N \times D_1 \times D_2 \times D_3 \times 3$ with entries $\mathbf{M}_{n,k_1,k_2,k_3,i} = k_i/D_i - 0.5$. Next, for each point we compute the corresponding Gaussian:

$$G_{n,k_1,k_2,k_3} = \exp(-0.5\sigma_n^{-2} \left\| \mathbf{M}_{n,k_1,k_2,k_3} - \mathbf{x}_n \right\|^2). \tag{1}$$

Finally, we sum these to get the resulting volume: $\mathbf{o}_{k_1,k_2,k_3} = \sum_{n=0}^{N-1} G_{n,k_1,k_2,k_3}$. This implementation is simple and allows for independently changing the sizes of points. However, on the downside, both memory and computation requirements scale linearly with the number of points.

Since linear scaling with the number of points makes large-scale experiments impractical, we implemented the `fast` version of the method that has lower computation and memory requirements. We implement the conversion procedure as a composition of trilinear interpolation and a convolution. Efficiency comes at the cost of using the same kernel for all points. We implemented trilinear interpolation using the Tensorflow `scatter_nd` function. We used standard 3D convolutions for the second step. For improved efficiency, we factorized them into three 1D convolutions along the three axes.

---

[*]Work done while interning at Intel.

|           | Full | No drop. | Fixed $\sigma = 1.6$ | Fixed scale | 4000 pts. | 2000 pts. | 1000 pts. |
|-----------|------|----------|----------------------|-------------|-----------|-----------|-----------|
| Precision | 2.05 | 2.60     | 2.06                 | 2.01        | 2.10      | 2.17      | 2.11      |
| Coverage  | 1.98 | 1.99     | 2.82                 | 2.26        | 2.19      | 2.54      | 2.98      |
| Chamfer   | 4.03 | 4.59     | 4.89                 | 4.27        | 4.28      | 4.70      | 5.10      |

Table 1: Ablation study of our method for shape prediction. We report the Chamfer distance between normalized point clouds, multiplied by 100, as well as precision and coverage.

### A.3 Quantitative evaluation

To extract a point cloud from the ground truth meshes, we used the vertex densification procedure of Lin et al. [1]. For the outputs of voxel-based methods, we extract the surface mesh with the marching cubes algorithm and sample roughly 10000 points from the computed surface. We tuned the threshold parameters of the marching cubes algorithm based on the Chamfer distance on the validation set.

For pose evaluations, we computed the angular difference between two rotations represented with quaternions $q_1$ and $q_2$ as $2 \arccos\left(q_1 q_2^{-1} / \left\| q_1 q_2^{-1} \right\|\right)$.

## B   Additional experiments

### B.1   Ablation study

We evaluate the effect of different components of the model on the shape prediction quality. We measure these by training with pose supervision on ShapeNet chairs, with $64^2$ resolution of the training images. Results are presented in Table 1. The "Full" method is trained with 8000 point, point dropout, sigma schedule, and learned point scale. All our techniques are useful, but generally the method is not too sensitive to these.

### B.2   Additional qualitative results

Additional qualitative results are shown in Figure 1. Note that for MVC we use the binarization threshold that led to the best quantitative results.

### B.3   Towards part-based models

In most experiments in the paper the shape parameters of the points were set by hand, and only the scaling factor was learned. However, our formulation allows learning the shape parameters jointly with the positions of the points. Here we explore this direction using the `basic` implementation, since it allows for learning a separate shape for each point in the point set. We explore two possibilities: isotropic Gaussians, parametrized by a single scalar and general covariance matrices, parametrized by 7 numbers: 3 diagonal values and a quaternion representing the rotation (this is an overcomplete representation). This resembles part-based models: now instead of composing the object of "atomic" points, a whole object part can be represented by a single Gaussian of appropriate shape (for instance, an elongated Gaussian can represent a leg of a chair).

Figure 2 qualitatively demonstrates the advantage of the more flexible model over the simpler alternative with isotropic Gaussians. One could imagine employing yet more general and flexible per-point shape models, and we see this as an exciting direction of future work.

Figure 3 shows the projection error of different approaches for varying number of points in the set. Learnable parameters perform better than hand-tuned and learned full covariance performs better than learned isotropic covariance. A caveat is that training with full covariance matrix is computationally more heavy in our implementation.

### B.4   Additional visualizations of semantic correspondences

Additional visualizations of semantic correspondences are shown in Figure 4. We use the same two templates here as in the main paper.

Figure 1: Additional qualitative results of shape prediction. Best viewed on screen.

Figure 2: Silhouettes learned with full learned Gaussian covariance versus hand-tuned isotropic Gaussian, using 20 points.

Figure 3: Projection error with different models and different number of points. More flexible density distributions allow for reaching the same error with fewer points. In particular, full learnable covariance can require roughly an order of magnitude fewer points than hand-tuned isotropic covariance to reach the same quality.

## B.5 Interpolation of shapes in the latent space

Fig. 5 shows results of linear interpolation between shapes in the latent space given by the first (shared) fully connected layer. We can observe gradual transitions between shapes, which indicates that the model learns a smooth representation of the shape space. Failure cases, such as legs in the second row, can be attributed to the limited representation of the office chairs with 5 legs in the dataset.

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

Figure 4: Additional visualizations of semantic correspondences.

Input A    Prediction A                                          Prediction B    Input B

Figure 5: Interpolation of shapes in the latent space.