[Reviews · NeurIPS 2018]

Reviewer 1



I maintain my original review and think the paper should be accepted. --------- The paper proposes to learn to predict both shape and pose of an object given a set of views of the object using multiview consistency. To get around the ambiguity of shape and pose, the authors propose to have an ensemble of pose predictors, which they distill post-training into a single model. I am inclined to accept the paper. The method is a solid solution to an interesting problem and the paper is well-written. In more detail: a) This is clearly a novel solution to an interesting but, so far, poorly explored problem. The entire unsupervised learning of shape from image collections makes a lot of sense, but most past work has relied on ground-truth pose, which is ecologically implausible. This work proposes a solution that indeed seems to work, and the only similar work that I am aware of is [20], which requires having some prior on the camera/object pose. This paper's proposed approach is cleaner, and so the paper tackles a timely and unsolved problem with a sensible solution that seems to work. I think this is an obvious accept. b) As a side benefit, the differential point cloud formulation is quite nice. I would be skeptical if the point clouds were isotropic (and few), but with a number of anisotropic points, the results look quite nice. My only concern would be getting out other forms of shapes (e.g., a mesh or voxel grid), since simply cutting at the isocontours might lead to weird surface normals, but all representations have their difficulties and that's probably fixable. c) The experiments seem quite thorough and the authors compared to all the relevant baselines and included sensible ablation and control experiments (including validating the proposed representation of point gaussians in space against voxels). d) Finally, I feel that the paper is well-written. I have a good sense of how the work differs from past work and what the contributions are, and I feel as if the authors are being clear about their method and fair to past work. One small quibble with the authors is the claim that voxels are unscalable. Naively, yes, but if one uses the frameworks from either Haene or Tatarchenko, one can produce high resolution voxels.

Reviewer 2



This paper presents a method for unsupervised learning of shape and pose with differentiable point clouds. During training, it uses multiple views of the same object as input, and estimate the camera poses and the 3d point cloud by minimizing the reprojection loss between the multiple input images. The key novelty is the differentiable point cloud rendering (reprojection) operation (Fig.2), which is similar to operations in CVPR 2018 SplatNet. Overall, the differentiable point cloud rendering is interesting, which potential applications to many other vision tasks. The experimental results are promising. A few questions. 1. If camera pose is unknown, even with multiple images, there is inherently ambiguity for 3D reconstruction. For example, from two views, we can obtain perspective reconstruction at best. When more information is present, for example, camera is calibration, we can then obtain affine reconstruction and metric reconstruction. Given this, I want to know how the paper handles such reconstruction ambiguity from a single input image? 2. What is the benefit of using a point cloud as the 3D representation, compared to 3D meshes, for example? There is a recent CVPR 2018 paper on nenural mesh rendering? Will that be used for the same task? If so, what is the pro and con? 3. The reprojection loss heavily depends on the background. What the algorithm will do, if the input images are real images with cluttered background?

Reviewer 3



This paper introduces an approach to learning to predict 3D shape with 2D images only as input. The core idea is to make use of a 3D points-based representation of shape and to additionally predict the pose of the observed object, so as not to require this information during training. In general, the paper is clearly written and does not suffer from any major flaw. My main concern is that I find the contribution of this paper a bit weak for NIPS. See detailed comments below. Novelty: - As acknowledged by the authors, there has been an increasing interest in learning to predict 3D shape from 2D information only. In this context, [6] and [11] also represent shape as a 3D point cloud. Furthermore, [13] and [20] also work in the setting where pose is not available during training. I find this to limit the contributions of this submission. - I acknowledge that there can be different approaches to solving the same problem, which could be the focus here. However, the proposed method does not seem to be of great originality, essentially defining a network that directly outputs 3D points and pose. I might be missing something, but currently the paper fails to convince me of the novelty of the method. Method: - Section 4 discusses the notion of differentiable point clouds, which is achieved by converting the point cloud to a volume. Considering that [6] and [11] have also worked with point cloud representations within deep networks, I do not see why this different strategy is necessary or beneficial. I would appreciate if the authors could clarify this. - When predicting a point cloud, the order of the points in the set is irrelevant. This suggests that there are ambiguities in what the network can predict. I was wondering if the authors have observed problems related to these ambiguities, or if at least they could discuss this issue. Experiments: - In Section 5.2, it would be interesting to compare with [11], which also uses a point-cloud representation and works with known pose. - In Section 5.3, it would be interesting to compare with [13], which also handles the unknown pose scenario. - For the classification metric, 30 degrees seems like a very large threshold. It would be worth evaluating what happens with smaller thresholds. - In practice, at test time, is there any reason to also predict the pose? It seems it is mostly the shape that is of interest. - In Section 5.3, the authors report results of Ours-basic and Ours, referred to as the full model. Is Ours what was described as "fast" in Section 4.1? If so, considering that it performs better than Ours-basic, is there any value in the basic algorithm? From Section 4.1, it is supposed to be more flexible, but the quantitative experiments don't really reflect this (although Fig. 5 briefly illustrates this). Related work: - It might be worth citing Cohen & Welling, 2014, "Transformation Properties of Learned Visual Representations", and Worrall et al., ICCV 2017, "Interpretable transformations with encoder-decoder networks". These methods also exploit a point-based representation but in the context of novel view synthesis. - Rhodin et al., CVPR 2018, "Learning Monocular 3D Human Pose Estimation From Multi-View Images" also make use of multiview data with unknown pose 3D human pose estimation.